

# *Kekveus brevisulcatus* sp. nov., a new featherwing beetle from mid-Cretaceous amber of northern Myanmar (Coleoptera: Ptiliidae)

Yan-Da Li[1,2], Shûhei Yamamoto[3,4], Alfred F. Newton[4] and Chen-Yang Cai[1,2]

[1] Bristol Palaeobiology Group, School of Earth Sciences, University of Bristol, Bristol, United Kingdom
[2] State Key Laboratory of Palaeobiology and Stratigraphy, Nanjing Institute of Geology and Palaeontology, Chinese Academy of Sciences, Nanjing, China
[3] The Hokkaido University Museum, Hokkaido University, Sapporo, Japan
[4] Negaunee Integrative Research Center, Field Museum of Natural History, Chicago, IL, United States of America

## ABSTRACT

Ptiliidae (featherwing beetles) is a group of minute staphylinoid beetles with a scarce fossil record. Here a second member of the Mesozoic genus *Kekveus* Yamamoto et al. is reported from mid-Cretaceous Burmese amber, with detailed morphology obtained through confocal microscopy. *Kekveus brevisulcatus* Li, Yamamoto, Newton & Cai sp. nov. shares with *K. jason* Yamamoto et al. the unpaired medial pronotal fovea and narrowly separated transverse metacoxae, but can be separated from the latter based on its less elongate body, shorter pronotal foveae, and much weaker transverse depression on the head. Our phylogenetic analyses support the discheramocephalin affinity of *Kekveus*, although its relationship with other members of Discheramocephalini cannot be confidently resolved.

## INTRODUCTION

Ptiliidae is a distinctive family of miniaturized staphylinoid beetles (*Polilov, 2016*; *Hall, 2016*), with nearly 1,000 species in 100 genera worldwide (*Newton, 2022*). The majority of ptiliids are less than 1 mm long (*Polilov et al., 2019a*), with the smallest reaching only 325 μm long (*Polilov, 2015*). To adapt for flight under miniaturization, the hind wings of ptiliids have been evolved to be feather-like, *i.e.,* with long setae on a narrow blade (*Polilov et al., 2019b*; *Farisenkov et al., 2022*). The phylogenetic position of Ptiliidae within Staphylinoidea is robustly resolved as sister to Hydraenidae (*Zhang et al., 2018*; *Cai et al., 2022*). The family presently contains two subfamilies: Nossidiinae and Ptiliinae (*Polilov et al., 2019a*). The more plesiomorphic Nossidiinae includes only four extant genera (*Sörensson & Delgado, 2019*), with an additional fossil genus recently reported from mid-Cretaceous Burmese amber (*Li et al., 2022a*). Ptiliinae is more diversified, with seven tribes currently recognized (*Polilov et al., 2019a*).

Corresponding author
Chen-Yang Cai, cycai@nigpas.ac.cn

Among the ptiliine tribes, Discheramocephalini was erected by *Grebennikov (2009)* to include several genera with horizontal cavities on the mesoventrite (*e.g.*, *Grebennikov & Leschen, 2010*: Figs. 15 and 17–19; *Darby, 2013*: Figs. 22 and 37). However, similar mesoventral fossae can also occur in genera outside of Discheramocephalini (*e.g.*, *Sindosium* Johnson and *Millidium* Motschulsky; *Polilov et al., 2019a*), and in some genera of Discheramocephalini the mesoventral fossae might be unclear (*Cissidium* Motschulsky; *Darby, 2015*; *Darby, 2019*) or absent (*Americoptilium* Darby; *Darby, 2018*; *Darby, 2020a*). According to *Polilov et al. (2019a)*, the monophyly of Discheramocephalini was not supported by analyses on either morphological or molecular datasets. Nevertheless, the tribe is provisionally retained due to insufficient evidence for an alternative arrangement (*Polilov et al., 2019a*).

*Yamamoto, Grebennikov & Takahashi (2018)* described the first fossil of Discheramocephalini, *Kekveus jason* Yamamoto et al. from Burmese amber. Based on the presence of clear grooves on the pronotum, *Kekveus* was suggested to be closely related to the genera *Skidmorella* Johnson and *Discheramocephalus* Johnson, although no phylogenetic analysis was performed to confirm this placement. In the present study, we describe a new member of *Kekveus* from Burmese amber, and attempt to evaluate its position phylogenetically based on the detailed morphology obtained with confocal microscopy.

## MATERIALS & METHODS

The research procedure is generally similar to that used in previous studies by the authors (*e.g.*, *Li et al., 2022a*; *Li et al., 2022b*; *Li et al., 2022c*; *Li et al., 2023*).

### Materials

The Burmese amber specimen studied herein originated from amber mines near Noije Bum (26°20′N, 96°36′E), Hukawng Valley, Kachin State, northern Myanmar, and is deposited in the Nanjing Institute of Geology and Palaeontology (NIGP), Chinese Academy of Sciences, Nanjing, China. Jewellery-grade Burmese amber specimens are commonly carried and sold legally in Ruili, Dehong Prefecture on the border between China and Myanmar. The specimen in this study was purchased in 2015 (prior to 2017), and was therefore not involved in the armed conflict and ethnic strife in Myanmar. The holotype of *Kekveus brevisulcatus* sp. nov. (NIGP200739-1) is preserved along with an euaesthetine staphylinid (NIGP200739-2) in the same amber piece.

For a comparative purpose, the second author (S.Y.) examined the holotype of *K. jason* also in Burmese amber, which is housed in the Gantz Family Collections Center (as 'Integrative Research Center' in *Yamamoto, Grebennikov & Takahashi, 2018*), Field Museum of Natural History (FMNH), Chicago, IL, USA, under the registered number FMNHINS-3741459.

### Imaging and description

Brightfield images were taken with a Zeiss Discovery V20 stereo microscope. Confocal images were obtained with a Zeiss LSM710 confocal laser scanning microscope, using the 488 nm Argon laser excitation line (*Fu et al., 2021*; *Li et al., 2023*). Brightfield images were

stacked with Helicon Focus 7.0.2. Confocal images were semi-manually stacked with ZEN 3.4 (Blue Edition) and Adobe Photoshop CC. Images were further processed in Adobe Photoshop CC to adjust brightness and contrast.

The general morphological terminology follows *Lawrence & Ślipiński (2013)*. The terminology for mesoventral structures follows *Darby (2020b)*.

## Phylogenetic analyses

We applied two matrices to evaluate the phylogenetic position of the new fossil (Data S1–S3). The matrix with broader sampling among the whole Ptiliidae was compiled by *Polilov et al. (2019a)*, with subsequent minor corrections by *Li et al. (2022a)*. The matrix focusing on Discheramocephalini was compiled by *Grebennikov (2009)*, with the definitions of some characters slightly modified to fit the inclusion of the new fossil. We successfully coded 34 of 68 characters in the first matrix, and 25 of 37 characters in the second matrix for the new fossil. Constrained analyses were performed under both Bayesian inference and weighted parsimony. The constraints were created based on the Bayesian molecular tree (their Figs. 8 and S9) by *Polilov et al. (2019a)*.

The Bayesian analyses were performed using MrBayes 3.2.6 (*Ronquist et al., 2012*). Two MCMC analyses were run simultaneously, each with one cold chain and three heated chains. Trees were sampled every 2,000 generations. Analyses were stopped when the average standard deviation of split frequencies remained below 0.01. The first 25% of sampled trees were discarded as burn-in, and the remainder were used to build a majority-rule consensus tree.

The parsimony analyses were performed under implied weights with R 4.1.0 (*R Core Team, 2021*) and the R package TreeSearch 1.0.1 (*Smith, 2023*). Parsimony analyses achieve highest accuracy under a moderate weighting scheme (*i.e.,* when concavity constants, K, are between 5 and 20) (*Goloboff, Torres & Arias, 2018*; *Smith, 2019*). Therefore, the concavity constant was set to 12 here, as suggested by *Goloboff, Torres & Arias (2018)*. For the analysis based on the matrix by *Grebennikov (2009)*, where fewer constraints were applied, clade supports were generated based on 5,000 jackknife pseudoreplicates.

The trees were drawn with the online tool iTOL 6.6 (*Letunic & Bork, 2021*) and graphically edited with Adobe Illustrator CC 2017.

## Nomenclature

The electronic version of this article in Portable Document Format (PDF) will represent a published work according to the International Commission on Zoological Nomenclature (ICZN), and hence the new name contained in the electronic version is effectively published under that Code from the electronic edition alone. This published work and the nomenclatural act it contains have been registered in ZooBank, the online registration system for the ICZN. The ZooBank LSIDs (Life Science Identifiers) can be resolved and the associated information viewed through any standard web browser by appending the LSID to the prefix http://zoobank.org/. The LSID for this publication is: EE933181-B8D2-40B5-B302-4B5EDFFCAFC1. The online version of this work is archived and available from the following digital repositories: PeerJ, PubMed Central SCIE and CLOCKSS.
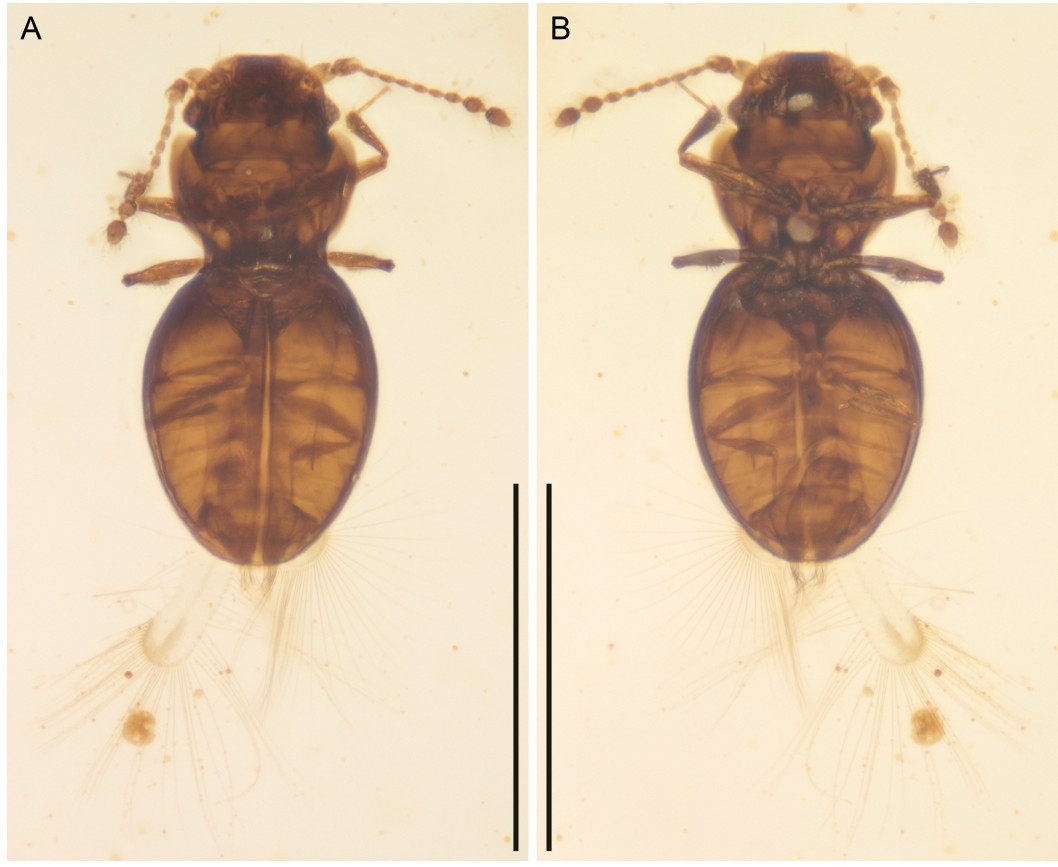

**Figure 1** *Kekveus brevisulcatus* **Li, Yamamoto, Newton & Cai sp. nov., holotype, NIGP200739-1, under brightfield microscopy.** (A) Habitus, dorsal view. (B) Habitus, ventral view. Scale bars: 400 μm.

## SYSTEMATIC PALEONTOLOGY

Order Coleoptera Linnaeus, 1758

Superfamily Staphylinoidea Latreille, 1802

Family Ptiliidae Erichson, 1845

Subfamily Ptiliinae Erichson, 1845

Tribe Discheramocephalini *Grebennikov, 2009*

Genus *Kekveus Yamamoto, Grebennikov & Takahashi, 2018*

### *Kekveus brevisulcatus* Li, Yamamoto, Newton & Cai sp. nov.

(Figs. 1 and 2)

**Material.** Holotype, probably male, NIGP200739-1.

**Etymology.** The specific name refers to its shorter foveae on the pronotal disc (compared with the type species *K. jason*).

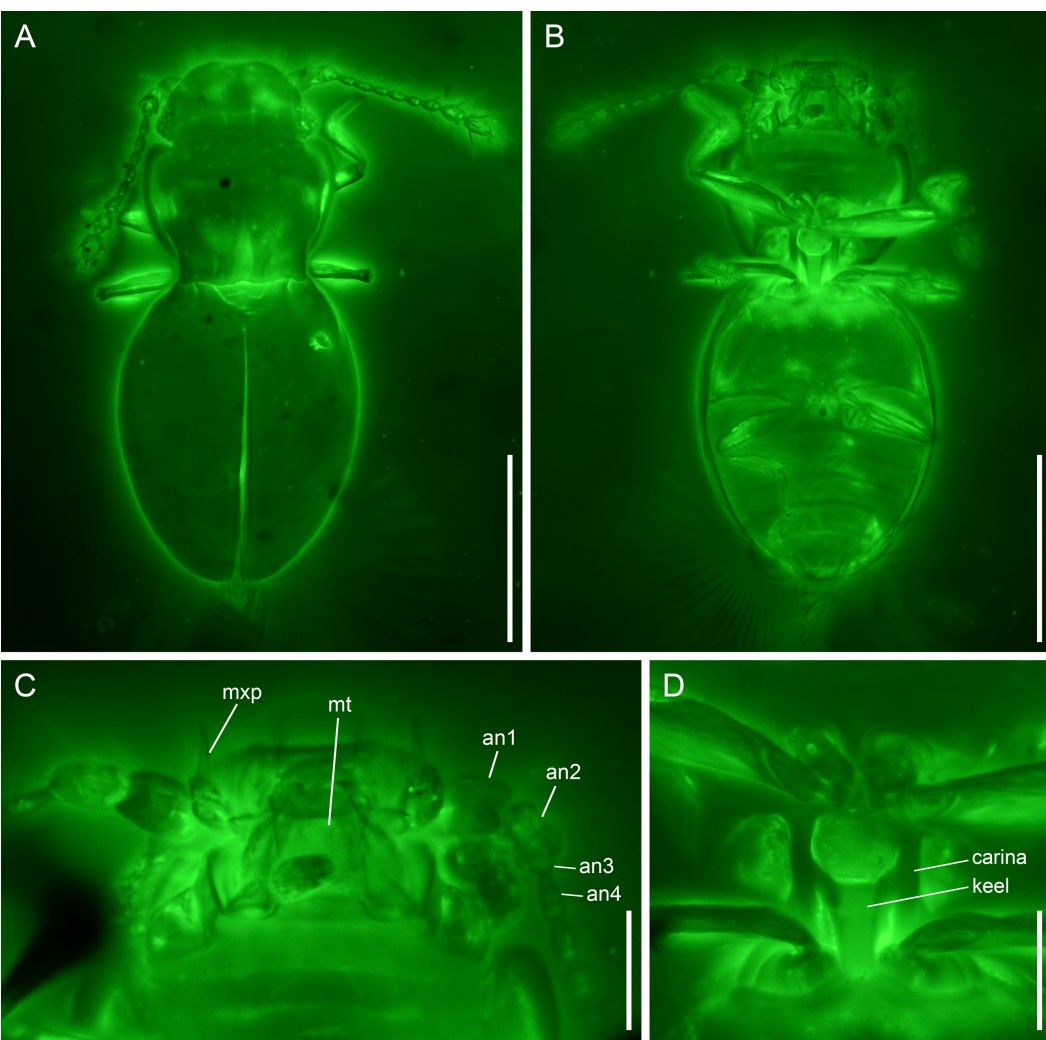

**Figure 2** *Kekveus brevisulcatus* Li, Yamamoto, Newton & Cai sp. nov., holotype, NIGP200739-1, **under confocal microscopy.** (A) Habitus, dorsal view. (B) Habitus, ventral view. (C) Mouthparts, ventral view. (D) Mesoventrite, ventral view. Abbreviations: an1–4, antennomeres 1–4; mt, mentum; mxp, maxillary palp. Scale bars: 200 µm in (A–B), 50 µm in (C–D).

**Locality and horizon.** Amber mine located near Noije Bum Village, Tanai Township, Myitkyina District, Kachin State, Myanmar; unnamed horizon, mid-Cretaceous, Upper Albian to Lower Cenomanian.

**Diagnosis.** Body moderately elongate. Head dorsally with only a weak transverse depression behind eyes (Fig. 2A). Pronotal disc with three foveae; medial fovea less than 1/2 pronotal length; lateral foveae less than 1/4 pronotal length (Fig. 2A). Mesoventrite as illustrated in Fig. 2D (see also Description). Metacoxae strongly transverse, narrowly separated (Fig. 2B).

**Description.** Body moderately elongate, flattened (height/width ratio <0.5), constricted at pronotal base in dorsal view, about 0.56 mm long, 0.26 mm wide.

Head with vertex somewhat declined; transverse depression behind eyes weak and simple. Compound eyes well-developed. Antennal grooves absent. Antennae 11-segmented, generally filiform; antennomeres 1–2 moderately enlarged; antennomeres 3–8 slender and elongate, with similar width; antennomeres 9–11 forming a loose club; antennomere 9 about $1.2\times$ as wide as 8; antennomeres 10–11 about $1.8\times$ as wide as 8; antennomere 11 not constricted at middle. Maxillary palps 4-segmented; palpomere 3 thicker than other segments, oval; palpomere 4 aciculate. Mentum subparallel-sided.

Pronotum widest in anterior half, narrowed at base; anterior angles slightly produced, rounded; disc with three foveae; medial fovea less than 1/2 pronotal length; lateral foveae less than 1/4 pronotal length. Prosternal process probably narrow. Procoxae (almost) touching apically.

Scutellar shield triangular, without fossae or longitudinal keel. Elytra completely covering abdomen; surface with a pair of foveae at base, otherwise glabrous. Mesoventrite without medial extension of collar; lateral carinae subparallel, extending from mesocoxal cavities to anterior mesoventral margin; keel anteriorly terminating truncately at middle of mesoventrite, with branches extending anterolaterally to connect with lateral carinae. Metacoxae narrowly separated. Meso-metaventral junction externally clearly visible. Metaventrite broad, without lines or impressions. Metacoxae strongly transverse, narrowly separated, without large plates.

Legs short and slender. Femora not strongly flattened. Tibiae weakly expanded distally, with relatively stout setae. Tarsi of typical ptiliid type, cylindrical. Pretarsal claws simple, equal in size.

Hind wings with moderately narrow blade and long setae.

Abdomen with seven sternites (sternites III–IX) exposed (in the holotype); surface without serration or cavities.

**Remarks.** The new species shares with *Kekveus jason* the general body shape (constricted at pronotal base, and relatively flat), unmodified antennal club, distinct fovea on elytral shoulder (fossa as described by *Yamamoto, Grebennikov & Takahashi, 2018*) and otherwise glabrous elytral surface, and more importantly, the unpaired medial longitudinal fovea on pronotum and narrowly separated transverse metacoxae (see also Discussion). Thus, it can be confidently placed in the genus *Kekveus*.

*Kekveus brevisulcatus* can be easily separated from *K. jason* based on its less elongate body and shorter foveae on pronotum. In *K. jason*, the medial pronotal fovea is distinctly longer than half of the pronotal length, and the lateral foveae also reach half of the pronotal length (*Yamamoto, Grebennikov & Takahashi, 2018*), while in *K. brevisulcatus*, the medial pronotal fovea does extend beyond half of the pronotal length, and the lateral foveae are less than 1/4 of the pronotal length. The transverse depression on the head of *K. brevisulcatus* is much weaker than that of *K. jason*, and is more reminiscent of what may be seen in some *Cissidium* or *Dacrysoma* Grebennikov (*e.g.*, *Grebennikov, 2009*: Fig. 8F; *Darby, 2013*: Fig. 58; *Darby, 2020b*: Fig. 3B). The mesoventral morphology is often used for species-level differentiation in discheramocephalins, but its state in *K. jason* is not very clear. The anteriorly branched mesoventral keel in *K. brevisulcatus* is quite rare in Discheramocephalini. A somewhat

similar branched keel may also be found in *Fenestellidium canfordanum* Darby, although the lateral carinae are absent in the latter (*Darby, 2011*).

Some other inconsistencies between *Yamamoto, Grebennikov & Takahashi (2018)* and the characters of *K. brevisulcatus* observed through confocal microscopy, however, raise questions about the validity of some character interpretation by *Yamamoto, Grebennikov & Takahashi (2018)*. They proposed that the fossae on the elytral shoulder in *Kekveus* might be homologous to the fossae on the scutellum found in some other discheramocephalins. Here the morphology of this concavity on the elytral shoulder is clearly imaged for *K. brevisulcatus*, and such foveae on the elytral shoulder may also be seen in extant discheramocephalins either with or without fossae on the scutellum (*e.g.*, *Darby, 2013*, Fig. 20; *Darby, 2020b*: Figs. 16B and 17B). Therefore, no homology should be assumed for these concavities occurring on different sclerites. *Yamamoto, Grebennikov & Takahashi (2018)* also indicated a possible presence of the horizontally-oriented mesoventral fossae in *K. jason*, which was suggested as a possible apomorphy of Discheramocephalini. However, as in many other discheramocephalins, the mesoventrite of *Kekveus* is distinctly elevated medially. Thus, even if such horizontal fossae are present, generally they would not be clearly visible in an exactly ventral view (*e.g.*, *Grebennikov, 2008*: Figs. 10 and 11; *Darby, 2013*: Figs. 21 and 22).

## DISCUSSION

*Kekveus* was first reported by *Yamamoto, Grebennikov & Takahashi (2018)*, with a single representative, *Kekveus jason*, from Burmese amber. As written by *Yamamoto, Grebennikov & Takahashi (2018)*, their attempt to evaluate the placement of *Kekveus* with a phylogenetic analysis failed, since many characters cannot be directly observed (under brightfield microscopy). Nevertheless, they assigned it to the tribe Discheramocephalini, mainly based on the grooves and fossae on its surface. In the present study, with the aid of confocal microscopy, we were able to observe more detailed morphology of *Kekveus brevisulcatus* sp. nov., which made it possible to assess the systematic position of *Kekveus* by formal phylogenetic analyses.

We first tested the placement of *Kekveus* within the whole Ptiliidae with the matrix developed by *Polilov et al. (2019a)*. In both Bayesian and parsimony analyses, *Kekveus* appeared to be sister to *Discheramocephalus* (Fig. 3, S1), which supports the discheramocephalin affinity of *Kekveus*, although the monophyly of the tribe Discheramocephalini itself remains dubious (*Polilov et al., 2019a*; *Sörensson & Delgado, 2019*). To further evaluate the relationships between *Kekveus* and other discheramocephalin genera, analyses were then conducted based on the matrix developed by *Grebennikov (2009)*. In the Bayesian analyses, *Kekveus* was nested in the basal polytomy of Ptiliinae (Fig. S2). In the more resolved parsimony result (*Smith, 2019*), *Kekveus* appeared to be associated with *Cissidium* and *Dacrysoma* (Fig. 4), rather than the group of *Discheramocephalus* and *Skidmorella* as originally supposed by *Yamamoto, Grebennikov & Takahashi (2018)*.

This inconsistency between the results from the two matrices is not totally unexpected, as *Kekveus* actually shares characters with both *Discheramocephalus* and *Cissidium* +

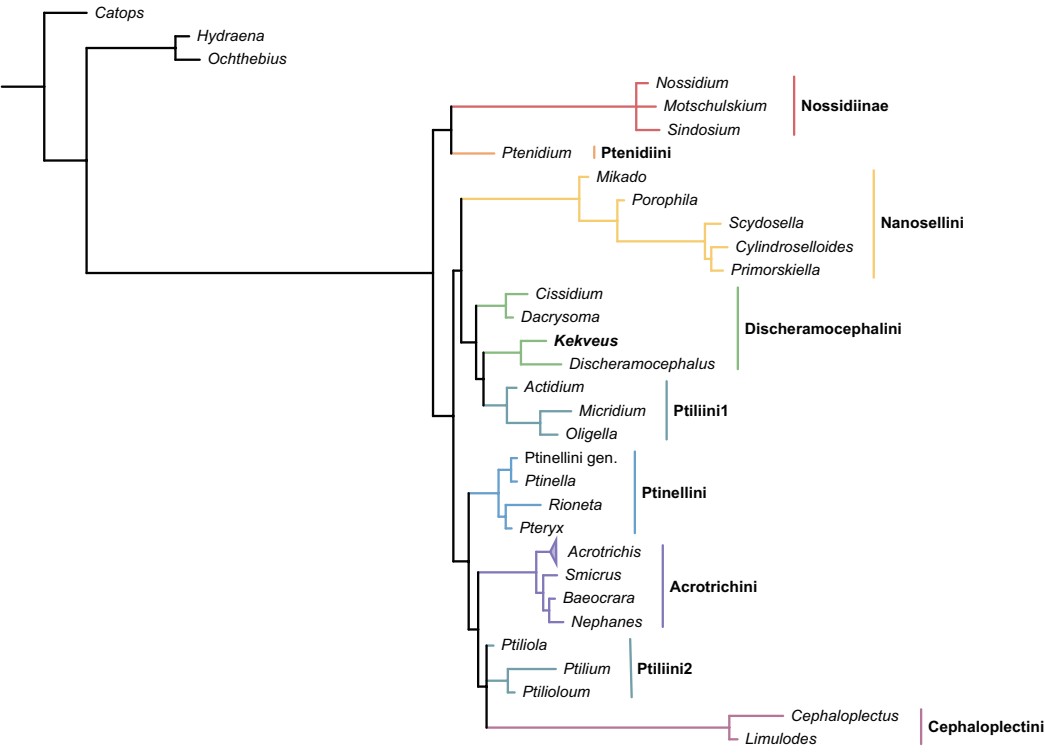

**Figure 3** **Placement of *Kekveus* within Ptiliidae, analyzed based on the matrix by *Polilov et al. (2019a)*.** Tree resulting from the constrained Bayesian analysis.

*Dacrysoma*. The narrowly separated transverse metacoxae is a key character voting for a relationship between *Kekveus* and *Discheramocephalus*. Such plesiomorphic metacoxae are characteristic for Nossidiinae, and are extremely rare in Ptiliinae, where the metacoxae are usually small and distinctly separated (*Polilov et al., 2019a*). Although in Nanosellini the metacoxae are also narrowly separated (*Sörensson, 1997*; *Hall, 1999*), they have wide metacoxal plates covering metafemora (*Polilov et al., 2019a*), which are absent in *Kekveus* and *Discheramocephalus*. *Kekveus* also shares with *Discheramocephalus*, as well as *Skidmorella*, the longitudinal pronotal foveae, although in the latter two genera the pronotal foveae are always paired (*Grebennikov, 2009*; *Darby, 2022*). Nevertheless, shorter but still longitudinally-oriented paired pronotal foveae may also be found in some *Cissidium* (*e.g.*, *Darby, 2020b*: Figs. 12B and 16B). The Ptiliini genera *Ptilium* Gyllenhal, *Numa* Darby, *Gomyella* Johnson and *Millidium* Motschulsky even have the unpaired medial longitudinal fovea on the pronotum (*Darby, 2020c*; *Darby, 2021*), suggesting this character might be quite variable and have limited value for higher-level relationships (*e.g.*, *Sawada & Hirowatari, 2003*). On the other hand, the structure of the meso-metaventral junction disfavors an affinity between *Kekveus* and *Discheramocephalus*. In *Discheramocephalus* and presumably related genera (*Fenestellidium* Grebennikov, *Africoptilium* Darby, *Americoptilium*, *Skidmorella*), the meso- and metaventrites are fused between the mesocoxae, without a clear suture (*Grebennikov, 2009*; *Darby, 2020a*). In

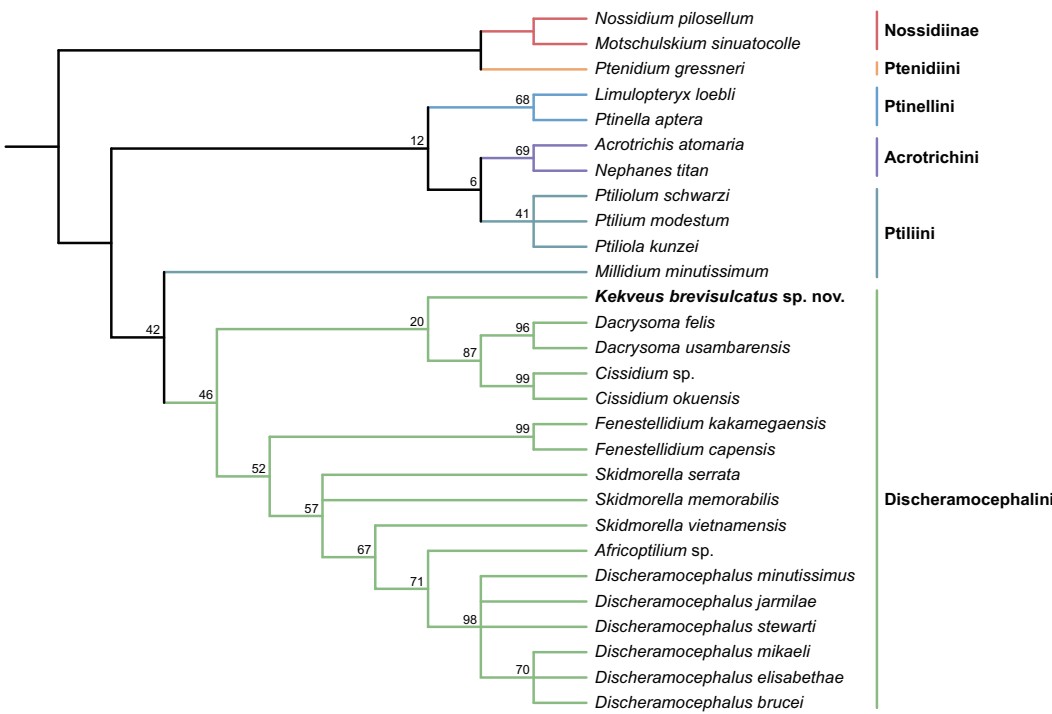

**Figure 4** **Relationship between *Kekveus brevisulcatus* and other Discheramocephalini, analyzed based on the matrix by *Grebennikov (2009)*.** Tree resulting from the constrained parsimony analysis under implied weights.

*Kekveus*, the mesoventral process ends near the posterior margin of the mesocoxae with a distinct suture, as seen in *Cissidium* and *Dacrysoma*, as well as many Ptiliini, Ptinellini and Acrotrichini (*Grebennikov, 2009*; *Polilov et al., 2019a*). *Discheramocephalus* additionally differs from *Kekveus* in the scutellum with a sharp keel (*e.g.*, *Darby, 2016*: Figs. 8–13) and abdominal sternite VIII with at least two deep cavities (*e.g.*, *Grebennikov, 2008*: Figs. 13 and 52; *Jałoszyński, 2020*: Figs. 7–9), and *Cissidium* and *Dacrysoma* additionally differ from *Kekveus* in having the apical antennomere constricted at middle (*e.g.*, *Darby, 2013*, Figs. 41–55; *Darby, 2015*: Figs. 20–25), although these are likely autapomorphies of the respective groups and cannot provide much information on the position of *Kekveus*.

Based on the above analyses and discussion, we show that the placement of *Kekveus* cannot be accurately resolved at the current stage. The monophyly of Discheramocephalini is contentious, and currently there is no densely sampled molecular phylogeny for the internal relationships of the tribe. A more comprehensive molecular phylogenetic framework for extant Discheramocephalini might be able to clarify the evolutionary trends of some morphological characters, which will be helpful to further determine the systematic position of *Kekveus*.

## ACKNOWLEDGEMENTS

We are grateful to Yan Fang for technical help with confocal imaging. Andrei Legalov and three anonymous reviewers provided helpful comments on the manuscript.

### Funding

Financial support was provided by the Strategic Priority Research Program of the Chinese Academy of Sciences (XDB26000000), the National Natural Science Foundation of China (42072022, 42222201, 42288201), and the Second Tibetan Plateau Scientific Expedition and Research project (2019QZKK0706). Yan-Da Li is supported by a scholarship granted by the China Scholarship Council (202108320010). Shûhei Yamamoto is supported by the Grant-in-Aid for JSPS Fellows (20J00159) from the Japan Society for the Promotion of Science (JSPS). The funders had no role in study design, data collection and analysis, decision to publish, or preparation of the manuscript.

### Grant Disclosures

The following grant information was disclosed by the authors:
Strategic Priority Research Program of the Chinese Academy of Sciences: XDB26000000.
National Natural Science Foundation of China: 42072022, 42222201, 42288201.
Second Tibetan Plateau Scientific Expedition and Research project: 2019QZKK0706.
China Scholarship Council: 202108320010.
Japan Society for the Promotion of Science: 20J00159.

### Competing Interests

The authors declare there are no competing interests.

### Author Contributions

- Yan-Da Li conceived and designed the experiments, performed the experiments, analyzed the data, prepared figures and/or tables, authored or reviewed drafts of the article, and approved the final draft.
- Shûhei Yamamoto and Alfred F. Newton analyzed the data, authored or reviewed drafts of the article, and approved the final draft.
- Chen-Yang Cai conceived and designed the experiments, performed the experiments, analyzed the data, authored or reviewed drafts of the article, and approved the final draft.

### Data Availability

The data for phylogenetic analyses are available in the Supplemental File.

The original confocal data are available in Zenodo: Li, Yan-Da, Yamamoto, Shûhei, Newton, Alfred F., & Cai, Chen-Yang. (2023). Confocal data of *Kekveus brevisulcatus*, holotype, NIGP200739-1 [Data set]. Zenodo. https://doi.org/10.5281/zenodo.7632980.

## New Species Registration

The following information was supplied regarding the registration of a newly described species:

Publication LSID: urn:lsid:zoobank.org:pub:EE933181-B8D2-40B5-B302-4B5EDFFCAFC1.

*Kekveus brevisulcatus* LSID: urn:lsid:zoobank.org:act:21CF3198-ED61-4D30-B5BC-E3668BC43EF1.

## Supplemental Information

Supplemental information for this article can be found online at http://dx.doi.org/10.7717/peerj.15306#supplemental-information.

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
