# Peer review of "Kekveus brevisulcatus sp. nov., a new featherwing beetle from mid-Cretaceous amber of northern Myanmar (Coleoptera: Ptiliidae)"

_PeerJ, doi:10.7717/peerj.15306_

## Round 0.1 · original submission · Minor Revisions

Your manuscript represents a significant contribution to the evolutionary history of an interesting family (Ptiliidae) from the Mesozoic era. However, the comments from reviewers 1 and 4 should be considered. I recommend minor revisions to be eligible to be published at the PeerJ.

·

Basic reporting

This is an interesting study that contributes to the knowledge of the Cretaceous biodiversity. The article is well formatted. The descriptions are correct and follow the Code of Zoological Nomenclature. The results given in the discussion are quite justified.
My comments:
References are mixed up in the work. The matrix and cladogram (Fig. 3) are given in Polilov et al. 2016b, not in Polilov et al. 2016a.
I did not find the matrix in the Grebennicov 2008.
Figure 4 to add the first species of the genus - Kekveus jason.
It is very important to discuss the time of origin and divergence of the group, based on data from molecular studies and paleontological data (including Paleogene) published by other researchers.

Experimental design

I think this section is ok.

Validity of the findings

I think this section is ok.

Reviewer 2 ·

Basic reporting

The article is devoted to an interesting group of beetles. Its structure, text and illustrations are of a high standard. All necessary literature is included. The article has a high degree of novelty, but the interest in its topic for a wide range of readers of an interdisciplinary journal is not obvious.

Experimental design

The article uses modern methods. It is a pity that there are no drawings in addition to photographs, since some of the important features cannot be seen in the photographs, but an attentive researcher could well see and draw them. It also seems from the photographs that the shape of the genitals can be seen on the confocal stacks, a description of which would be extremely useful.

Validity of the findings

The work is based on original material. Descriptions of the new taxon are made taking into account modern requirements. However, taxonomic interpretations do not seem convincing. It seems doubtful that this specimen belongs to the genus Kekveus, since the shape of the body and its parts, the location of the coxae, and the structure of the integument differ significantly from the description of the type species by Yamamoto et al. As in the original description of the genus, so in the description of the species, this article lacks important features, for example, details of the structure of the metathorax.

Reviewer 3 ·

Basic reporting

This paper represents a significant contribution towards our knowledge of Ptiliidae evolutionary history and systematics. The taxon identification is accurate, literature cited is sufficient, table/figures are well represented, and overall content of the is concise and well-supported.

I recommend publishing the paper.

Experimental design

Primary research falls within the scope of the journal.

Validity of the findings

no comment

Reviewer 4 ·

Basic reporting

Review report of manuscript #79143

”Kekveus brevisulcatus sp. nov., a new featherwing beetle from mid-Cretaceous amber of northern Myanmar (Coleoptera: Ptiliidae)” by Li, Yamamoto, Newton & Cai

1. BASIC REPORTING
English language is good. The following issues should be considered.

Add an ‘-e’ ending to ‘Discheramocephalin’ throughout the manuscript (many instances) when used adjectively (= Discheramocephaline)

Abstract: add authors to the scientific names of Kekveus and Kekveus jason respectively for clarity (and in order to fulfill the recommendations of the ICZN Code).

Line 53, first word: replace ‘discover’ with ‘describe’.

Line 75: delete ‘Mesoventrite as illustrated in Fig. 2D’ (or move to ‘Description’) since no diagnostic information is stated in that sentence (leaving to the reader to decide that herself), making it meaningless. Alternatively, explain in words (and arrows) what structures are diagnostic in Fig. 2D.

Line 139: replace ‘with’ with ‘by a’.

Line 140: replace: ‘in the whole of’ with ‘within’.

Line 152: replace ‘voting for’ with ‘speaking for’.

Line 207-8, meaning unclear: rephrase the sentence ‘The matrix on the whole Ptiliidae…’.

Line 244: add ‘.’ after the last ‘S’.

Line 249: journal name should be in italics.

Line 307: ‘In:’ should be in italics.

In Fig. 4 and in S2: the correct spelling is ‘Millidium minutissimum’.

Data S1, figure legend and many locations in text: wrong reference! Check, and where appropriate, cange to Polilov et al. (2019b).

Experimental design

Review report of manuscript #79143

”Kekveus brevisulcatus sp. nov., a new featherwing beetle from mid-Cretaceous amber of northern Myanmar (Coleoptera: Ptiliidae)” by Li, Yamamoto, Newton & Cai

2. EXPERIMENTAL DESIGN
The research presented is original, relevant and meaningful and amounts to a high technical standard. The methodology is described in sufficient detail for subsequent reproduction,

Validity of the findings

Review report of manuscript #79143

”Kekveus brevisulcatus sp. nov., a new featherwing beetle from mid-Cretaceous amber of northern Myanmar (Coleoptera: Ptiliidae)” by Li, Yamamoto, Newton & Cai

3. VALIDITY OF THE FINDING
The validity of the findings seems sound, but it should be pointed out that only 50% [34] of all available characters [68] extracted from Polilov (2019b) were coded for by Kekveus. This circumstance might deserve a comment. The high amount of question marks in the morphological matrix, i.e. as presented and used in S3, probably indicates a higher degree of systematic and phylogenetic placement uncertainty of Kekveus within the trees (Fig. 3 and S1) then accounted for under section ‘Discussion’ and as shown in the trees. If true, this should typically also be reflected by modest branch support values. Unfortunately, these were omitted in Fig. 3 and S1, respectively.

Additional comments

Review report of manuscript #79143

”Kekveus brevisulcatus sp. nov., a new featherwing beetle from mid-Cretaceous amber of northern Myanmar (Coleoptera: Ptiliidae)” by Li, Yamamoto, Newton & Cai

4. ADDITIONAL COMMENTS
This contribution is highly interesting from a scientific point of view but needs further consideration, including rectification of minor issues (see 1 above) and potential comments on phylogenetic robustness (of trees) and (lack of some) branch support values. In addition, the part under section ‘Discussion’ discussing distribution of various characters between different genera is somewhat messy and difficult to comprehend. It would need some ordered strengthening, possibly via a condensed, simple table lining up critical characters showing their presence/absence in the genera discussed in the text (lines 150-175).

---

## Round 0.2 · Minor Revisions

Dear Dr. Li

After a careful review of your re-submitted research, I confirm that you have responded to all the recommendations proposed by the reviewers, Therefore, your manuscript is ready now to be published at the PeerJ. Once again congratulation.

One minor point you need to address before final acceptance is that although you provide support that the fossils were acquired in 2015, you do not explicitly state this in the manuscript. Given the controversial nature of some of the amber material from Myanmar, this is necessary.

The paper would be ready for acceptance if the authors explicitly state in the manuscript something along the following lines in addition to storage of specimens in publicly-accessible collections: "The studied material was acquired in full compliance with the laws of Myanmar and China (work on this manuscript began in year XXXX). All authors declare that to the best of their knowledge, the fossils reported in this study were not involved in armed conflict and ethnic strife in Myanmar, and were acquired in 2015 (prior to 2017)."

Reviewer 4 ·

Basic reporting

No comment.

Experimental design

No comment.

Validity of the findings

No comment.

Additional comments

Second review report of manuscript #79143 by Reviewer #4 (anonymous).
”Kekveus brevisulcatus sp. nov., a new featherwing beetle from mid-Cretaceous amber of northern Myanmar (Coleoptera: Ptiliidae)” by Li, Yamamoto, Newton & Cai

The author’s revision and new version of manuscript #79143 rectifies previous typos and smaller mistakes, enhances general understanding and improves the manuscript content well, also including the supplementary material.

I recommend manuscript #79143 for publication.

---

## Round 0.3 · Minor Revisions

Dear authors,

Although you provide support that the fossils used in this study are acquired in 2015, you should explicitly declare this in the manuscript.

Best regards.

---

## Round 0.4 · accepted · Accept

Dear authors,

Thank you for addressing all of the reviewers' comments and for explicitly declaring in the material section that the fossils used in this study were from 2015.

Congratulations